# Next Generation Sequencing Approaches to Characterize the Respiratory Tract Virome

**DOI:** 10.3390/microorganisms10122327

**Published:** 2022-11-24

**Authors:** Nurlan Sandybayev, Vyacheslav Beloussov, Vitaliy Strochkov, Maxim Solomadin, Joanna Granica, Sergey Yegorov

**Affiliations:** 1Kazakhstan-Japan Innovation Center, Kazakh National Agrarian Research University, Almaty 050010, Kazakhstan; 2Molecular Genetics Laboratory TreeGene, Almaty 050009, Kazakhstan; 3School of Pharmacy, Karaganda Medical University, Karaganda 100000, Kazakhstan; 4Michael G. DeGroote Institute for Infectious Disease Research, Faculty of Health Sciences, McMaster University, Hamilton, ON L8S 4LB, Canada

**Keywords:** metagenomics, virome, DNA/RNA, virus, acute respiratory infections, ARVI, NGS

## Abstract

The COVID-19 pandemic and heightened perception of the risk of emerging viral infections have boosted the efforts to better understand the virome or complete repertoire of viruses in health and disease, with a focus on infectious respiratory diseases. Next-generation sequencing (NGS) is widely used to study microorganisms, allowing the elucidation of bacteria and viruses inhabiting different body systems and identifying new pathogens. However, NGS studies suffer from a lack of standardization, in particular, due to various methodological approaches and no single format for processing the results. Here, we review the main methodological approaches and key stages for studies of the human virome, with an emphasis on virome changes during acute respiratory viral infection, with applications for clinical diagnostics and epidemiologic analyses.

## 1. Introduction

Viruses are the most numerous organisms on Earth and the main driving forces of evolution, living in almost all ecosystems, including aquatic, terrestrial and symbiotic environments, at all levels of organization, in which they can exceed the number of bacteria by more than 10 times [1,2]. They also represent the largest variety of types of genome organization [3], which indicates not only different evolutionary origins, but also different interactions with the host. Viruses are of the greatest interest as infectious agents that regularly cause mass diseases, effectively avoid the immune defense mechanisms of the body, and overcome interspecies barriers, which is determined by the high genetic variability of viruses, which is especially characteristic of single-stranded RNA genomes [4].

The history of human virology marks the discovery of pathogenic viruses as ultra-filtered infectious particles capable of reproducing in cultured cells. The second wave of discoveries came with the development of molecular methods, which made it possible to detect major human viral pathogens that cannot be cultivated in vitro (for example, hepatitis C virus, HCV, hepatitis E virus, HEV, human herpesvirus, HHV8 [5].

With the discovery and the first use of massive parallel next-generation sequencing (NGS) methods that do not depend on the sequence of the target under study, began to appear related to a comprehensive study of the presence of DNA/RNA nucleic acids in various environmental objects and living organisms. For example, in 1998, to describe the totality of the genomes of microorganisms present in the environment, the term “metagenome” was introduced [6].

The first studies on the use of NGS were conducted by B. Dickins (2009) to study evolutionary changes in the DNA of the phage UX174. A high degree of polymorphism in the phage genome was shown, including in cultures grown for several hours. The results indicated the need for more studies using different platforms to differentiate between instrument errors and mutational flicker data. This metagenomics approach has become the conceptual basis for further research into other viral infections [7].

Examples of the practical application of methods for studying the metagenome are pathogen detection, species characterization, subtype identification, antimicrobial resistance detection, virulence profiling, microbiome composition [8], and factors affecting health and morbidity [9,10,11,12]. An example of the use of clinical samples for direct sequencing has an application for the rapid investigation and dissemination of information about severe acute respiratory syndrome coronavirus 2 (SARS-CoV-2), which causes COVID-19 [12].

The COVID-19 pandemic and possible future threats highlight the need for active study of the viral composition in various respiratory diseases [13]. The effective detection of human respiratory viral pathogens is critical in the investigation of patients with acute respiratory tract infections. The pandemic is likely to have a strong impact on the approaches and methods of integrated human virome research, especially on zoonotic infections. In this review, we consider the advantages and current state of metagenomic methods for virus identification and characterization, including all stages of research, as well as data on the virome in SARS, including COVID-19.

## 2. Human Virome

The human body contains various microorganisms, including viruses, bacteria, archaea, fungi, and protozoa. The human virome includes the collection of all viruses in the body, including bacteriophages, eukaryotic viruses, and endogenous retroviruses. Viruses are present throughout the human body, in the intestines, skin, and mouth, and can be detected in various types of samples, including blood, feces, cerebrospinal fluid, respiratory tract, and others [14]. For example, the most studied intestinal virome is the collection of all viruses that inhabit and coexist in the intestine, and are closely integrated with the bacterial microbiome, fungi and, other microbial communities that make up the microbiome [14,15].

Additionally, the human genome contains genetic elements of viruses (retroviral elements in the human genome and prophages in bacterial genomes). Viruses can be found on the surface of mucous membranes, and they often persist in other cell types, for example, in chronic herpesvirus infection in neuronal cells [15]. It was estimated that, in addition to integrated chromosomal viruses, every single healthy person is affected by more than ten persistent chronic eukaryotic viral infections that cause continuous activation of the immune system [16]. The most common are herpesviruses, polyomaviruses, anelloviruses, adenoviruses, papillomaviruses, and, for many people, additional viruses such as hepatitis B virus, hepatitis C virus, and HIV.

The development of metagenomic NGS has made it possible to study viral genomes in various clinical samples, including human feces [17,18,19], blood [20,21], cerebrospinal fluid [22,23], human tissues [24,25] and airway samples [26,27,28,29,30]. However, metagenomic sequencing of virus-like particles shows that only 14–87% of nucleotide sequences can be classified, which suggests that many uncharacterized viruses are still not studied [31]. An example of this is the discovery of the bacteriophage crAssphage, which is ubiquitous in human feces [32].

Interactions between phages, eukaryotic viruses, bacteria, and the individual’s immune system are likely central to host immune homeostasis. Eukaryotic viruses can cause acute or chronic infections, but they can also protect the host by triggering innate and/or adaptive immunity. Phages produced by bacteria can be taken up by immune cells and activate immune responses through TLR signaling. Bacteriophages can also change the abundance of bacterial species by lysing them, changing bacterial virulence, and inducing bacterial phagocytosis [14].

### Human Respiratory Virome

After the gut microbiome, the human oral microbiome is the second largest microbial community [14]. Like the gut virome, the oral virome is highly personalized and stable over time [33]. However, metagenomic analysis shows a much higher proportion of nucleotide sequences of bacteriophages in the respiratory virome compared to eukaryotic viruses, as well as a high proportion of unidentified sequences [21]. Additionally, some studies show that phage communities, unlike bacteria, have a greater diversity than in the intestine, which belongs to such families as Siphoviridae, Myoviridae, and Podoviridae. Interesting observations show that oral viromes are more similar in people with similar diets or oral bacterial communities, and people living in the same family, and the oral cavity is a heterogeneous compartment, with significant differences in the phage composition in saliva, plaque, and gums, which gives the idea and suggests that phage communities specific to a certain area of the oral cavity can model microbial dynamics in the oral cavity [21,33].

Concerning the content of eukaryotic viruses in the oral cavity and respiratory tract, they differ from each other, which may be due either to natural variations or differences in sample preparation and bioinformatic analysis. Analysis of the virome of healthy people revealed such viruses as cytomegalovirus, lymphocryptovirus, roseolovirus, herpes simplex virus, papillomaviruses, including unclassified, polyomavirus, mastadenovirus, dependovirus, alphatorkevirus, and unclassified anelloviruses [34]. Overall, these observations suggest that eukaryotic viruses are found in different parts of the oral cavity and respiratory tract, with much greater abundance and diversity in the nasal region [21].

A review of viruses found in the human respiratory tract under various pathological conditions [35] included picornaviruses—rhinoviruses A, B, and/or C [36], enteroviruses [37], parechovirus [38]; paramyxoviruses—respiratory syncytial virus [37], parainfluenza viruses 1–4 [39], metapneumovirus [40], measles virus [41], pneumovirus [38]; orthomyxoviruses—influenza virus A, B and/or C [40]; coronaviruses—HKU1, OC43, 229E and/or NL63 [42], adenoviruses [37] and many others.

From the time of SARS-CoV-2 appearance, scientific interest has shifted to the metagenome study, including viromes, in various forms of this disease. Thus, recent studies show that in patients with COVID-19, the virome was enriched in eukaryotic viruses, as well as in Escherichia and Enterobacter phages, which are associated with inflammatory processes in the intestine and the host’s interferon response; as well as increased expression of genes associated with stress, inflammation, and virulence, suggesting that these viruses play a role in the host’s immune response to SARS-CoV-2 infection [43].

Sequencing of the respiratory virome of patients with COVID-19 showed that 8% of samples co-infected with rhinovirus and influenza virus, which once again emphasizes the importance of a comprehensive study of the viral composition in ARVI [44]. Another study showed that viruses from the families Anelloviridae and Redondoviridae are more frequent colonizers and have higher titers in severe diseases [14].

## 3. ARVI as a Specific Case of Respiratory Virome Pathology

Acute respiratory viral infection (ARVI) is a general term for clinically and morphologically similar acute inflammatory diseases of the respiratory tract and lungs caused by viruses with tropism to the ciliated epithelium of the respiratory tract.

ARVI is the leading cause of morbidity and mortality worldwide [45,46,47,48,49,50]. According to WHO data, ARVI is the leading of all human infectious diseases, accounting for 90–95% of all cases and to up to 650,000 deaths per year [51]. Respiratory viral infections are quite easily transmitted from person to person and cause global pandemics with a certain frequency, as evidenced by the ongoing COVID-19 pandemic, which, as of 06.2022, has led to the infection of 539 million people worldwide and 6.3 million deaths [52].

In Kazakhstan were registered annually from 600,000 to 1 million cases of acute respiratory viral infections and COVID-19 infected more than 1.4 million people, of which more than 19,000 died [53].

Human respiratory viruses include various viruses (more than 200 species) that infect cells in the respiratory tract. They belong to different families and differ in structure, disease symptoms, circulation seasonality, transmissibility, and modes of transmission [50]. Many viruses are of zoonotic origin [54], as evidenced by the COVID-19 pandemic caused by SARS-CoV-2 [55].

Despite the developed service of laboratory diagnostics, a significant part of ARVI has an unidentified etiology, which complicates the clinical management of patients and the prediction of spread. According to various estimates, the proportion of such undetected pathogens can be 20–60% [40,56].

The most important etiological agents of severe human lower respiratory tract infections are bacteria such as *Streptococcus pneumoniae* and *Haemophilus influenzae*, as well as viruses such as respiratory syncytial virus (RSV), influenza virus, and SARS-CoV-2.

Viruses are more important in mild infections of the upper and middle respiratory tract and bronchitis in children, whereas bacteria are the main cause of pneumonia, especially in adults. Despite this, respiratory tract infections are often treated with antibiotics, which are ineffective in treating viral infections and lead to widespread antibiotic resistance. Systematic reviews and meta-analyses of the literature showed that 25% of adult patients with community-acquired pneumonia were viral. This percentage is higher in children, especially children under 5 years of age, due to the large proportion of cases caused by RSV [49].

The clinical symptoms of bacterial and viral infections are very similar and there is increasing evidence of mixed bacterial–viral infections and bacterial pneumonia existence, which are secondary to ARVI [49,57,58,59,60,61].

Most viruses are transmitted from person to person through the respiratory transmission route. Patients are most contagious in the early stages of the disease. Relapse rates can be particular high in semi-closed populations such as schoolchildren, inpatients, and nursing home residents. Children play an important role in family and community outbreaks of respiratory viruses. Frequent hand washing and covering the mouth when coughing or sneezing may partially prevent transmission [49].

Respiratory viruses belonging to different families differ in the degree of distribution and modes of transmission. Transmissibility, as measured by the basal reproduction number, is not uniform for the same virus. Respiratory viruses can be transmitted in four main ways: direct (physical) contact, indirect contact (fomites), (large) droplets, and (small) aerosols (Figure 1). Little is known about the relative contribution of each regimen to the transmission of a particular virus under different conditions and how its variations affect transmissibility and transmission dynamics. Understanding the relative contributions of different modes of transmission is critical to effective control and prevention strategies [50].

The diagnosis of ARVI includes clinical and laboratory approaches. Most often, the doctor diagnoses acute respiratory viral infections based on the patient’s complaints, visual examination of the nasopharynx, listening to the lungs, measurement body temperature, in addition, a general blood and urine test may be prescribed and, in some cases, if pneumonia is suspected, making X-ray examinations.

Laboratory diagnosis of ARVI includes the following approaches:Direct diagnostic methods for direct examination of biological material for the presence of a virus and/or viral antigen: electron microscopy (EM), enzyme immunoassay (ELISA), immunofluorescence reaction (RIF), radioimmunoassay (RIA), cytological methods;Direct molecular diagnostic methods for the presence of nucleic acids of viruses, based on DNA or RNA: PCR, RT-PCR, Real-Time PCR, sequencing, methods based on hybridization;Isolation and identification of the virus from clinical material in cell cultures;Serological diagnostics based on the determination of viral antibodies: complement fixation test (CFT), passive hemagglutination assay (PHA), indirect hemagglutination assay (IHA), and hemagglutination inhibition test (HI test).

Although the clinical examination of the patient makes it possible to establish the diagnosis, the syndromes of various infectious agents sometimes overlap and may be nonspecific. An etiological diagnosis can only be made with certainty by detecting viruses, antigens, or nucleic acids in respiratory or other samples, or retrospectively, by demonstrating an immune response in paired serum samples.

Virus isolation is still considered the gold standard, this method is laborious, rather time-consuming, and has low sensitivity. Simultaneously, molecular methods for detecting viral nucleic acids are fast and sensitive, and when used in a multiplex format, can detect the most common respiratory viruses. However, they are quite expensive and require expensive equipment, laboratory infrastructure, and well-trained personnel to operate properly, which limits their use under normal conditions [49,62].

Currently, there are two approaches to monitoring the spread and disease course of respiratory viruses, these are the measurement of plasma antibodies produced in response to viruses (IgG and IgM), which reflect the functionality of the host’s immune system, and the determination of viral load in respiratory samples, which indicates on virus replication in the focus of primary infection in humans [63,64,65].

## 4. NGS in the Human Respiratory Virome Study

Along with classical diagnostic methods such as virus isolation, serology, and PCR, NGS plays an important role in virus identification, especially in outbreaks of known and/or new diseases [66].

NGS technology is currently revolutionizing the field of genomics and clinical virology is no exception. High-throughput sequencing techniques have made significant contributions to many areas of virology, including virus discovery and metagenomics, molecular epidemiology, pathogenesis, and research into how viruses evade the host’s immune response. Previously, unknown viruses have been identified using NGS techniques, including a new rhabdovirus associated with acute hemorrhagic fever identified in Central Africa [67] and a new cyclovirus found in the cerebrospinal fluid of patients with infections of the central nervous system [68].

Metagenomics refers to the study of the complete genomic composition of a complex mixture of microorganisms [69]. Unlike bacteria, viruses do not have a common gene for all families, and therefore the study of the virome is based on complex analytical methods. In addition to detecting viruses, NGS is also capable of providing additional information on virulence markers, epidemiology, genotyping, and evolution of pathogens, as well as estimating the copy number from the number of DNA/RNA reads [69,70].

NGS methods are also used to study the genetic and phenotypic heterogeneity of viruses during replication in host cells, which is missed in conventional studies using consensus sequencing data. An example of such a study is Ikuyo Takayama (2021), who studied the genetic diversity of the A(H1N1)pdm09 virus using NGS in upper and lower respiratory tract samples from nine patients. Significant genetic heterogeneity was found in the samples, including 47 amino acid substitutions and 1 D222G/N substitution in the hemagglutinin was common to several patients. The authors note the need for such studies in order to not miss the potentially important mutations that occur during viral replication in the host, especially in patients with severe disease [71]. The NGS process typically consists of two parts. The first was experimental work in the laboratory (wet lab), including the stages of sample preparation, DNA/RNA extraction, library preparation, and sequencing. The second part is bioinformatic analysis (dry lab), which includes data quality control, removal of non-target DNA, and analysis of nucleotide sequences [72].

### 4.1. NGS Sequencing Methods

The application of NGS to viral studies has certain experimental and analytical features, in contrast to the study of microbial communities.

This applies to sample preparation and sequencing. For example, it must be considered that viral genomes (especially RNA) are rather fragile and easily destroyed and the ratio of viral material to the host genome is very low (less than 1%). Therefore, an important procedure is to conduct the procedure of amplification of target viral nucleic acids or enrichment of viral particles [4,69].

In some cases, it is necessary to reverse-transcribe the viral RNA before PCR and sequencing. PCR amplification leads to errors that are difficult to distinguish from real mutations. In addition to PCR errors, all NGS platforms introduce sequencing errors at a rate similar to the mutation rate in RNA viruses [73]. Additionally, the depth of sequencing, that is the number of unique reads including a given nucleotide in the sequence, can also vary depending on the genome [4].

Modern viral NGS protocols have already been optimized for detecting both RNA and DNA viruses [24,74,75]. In addition, viral particle enrichment techniques are often used to increase the relative concentration of viral particles and/or their nucleotide sequences, as well as methods of depleting host genomic DNA and ribosomal or mRNA. These methods are laborious and not easy to automate for routine use in clinical diagnostics, which imposes restrictions on their use in mass clinical diagnostics [69].

Difficulties in the detection of viruses by NGS in clinical specimens, especially respiratory ones, are due to the presence of an extremely small number of viruses and their nucleic acids in the study samples, compared with the high content of host genomic material and the bacterial component. These circumstances determine the high importance of the preliminary steps in NGS sequencing. Proper sampling and production of nucleic sequences of interest are critical for obtaining the desired results (Figure 2).

#### 4.1.1. Sample Collection

One great importance in the study of respiratory infections is the correct approach for sampling. Here, we need to know which of the viruses we have to detect. At the same time, an important point in the detection of respiratory viruses is the sampling and age of patients.

Respiratory viruses have different places of localization, as well as different times of incubation. For each virus, it is important to select its sampling scheme. As a sample, it can be suctioned, swabbed, washed, aspirated, and any sampling sites—nose, nasopharynx, throat, bronchoalveolar, sputum, feces, cerebrospinal fluid, or blood.

Sputum samples are more often used for the investigation of respiratory infections. However, some viruses penetrate deep into the respiratory system and require special methods such as aspirate collection [76].

Comparative analysis of nasopharyngeal swab (NFS) and nasopharyngeal aspirate (NPA) methods collected from 455 children, the sensitivity of NFS to the respiratory syncytial virus, influenza A virus, parainfluenza, and adenovirus was 98.4%, 100%, 100%, and 88.9%, respectively [77]. NPA used to isolate respiratory viruses in children was not effective when tested for respiratory pathogens in adults [78].

To detect influenza A and B viruses among 122 children, three sampling methods were tested using NPS, NPA, and nasal swabs (NS). Influenza A and B viruses were shown to be found in 85% of NPS, 78% of NS, and 69% of NPA [78].

In their studies, Gruteke (2004) showed the advantage of NPA for detecting respiratory viruses by PCR, demonstrating a diagnostic rate of 84% with NPA versus 58% when only transnasal or oropharyngeal swabs (OPS) were available [79].

High sensitivity is shown for the detection of respiratory viruses using NPA and NS samples for respiratory virus detection using culture, immunofluorescence, and multiplex PCR. However, when using PCR, the sensitivity obtained NPA was significantly higher than that with NS only for RSV [80].

The ability of viruses to accumulate in different parts of the body implies the right choice of methodology. Special swabs are also used to take samples for respiratory viruses, which allow deep penetration into the nasopharynx. For respiratory syncytial virus, rhinovirus, coronavirus, bocavirus, and enterovirus, nasopharyngeal swab collection is a suitable method, which may be suitable for PCR studies, virus isolation, and sequencing [81,82,83].

Carlo Ferravante (2022) used nasopharyngeal swabs of patients from the Campania region (Southern Italy) in the study of respiratory virome profiles against the background of SARS-CoV-2 infection [84]. Nasopharyngeal samples were used to detect respiratory viruses including IFVA, IFVB, PIV, RSV, hMPV, and hAdV [82]. Terlizzi M. (2009) used bronchoalveolar lavage as a sample when developing quantitative PCR to detect parainfluenza virus. [85].

There are also data suggesting the pooling of NS and OPS samples from the same patient in the same viral transport medium [86,87]. During a pandemic, it is recommended that each nasal and throat swab collected from adults and pooled to detect the influenza virus [88].

Detection of the influenza virus, using two detection methods, was tested on four different samples (NPS, OPS, nasal aspirate, sputum), it turned out that sputum and nasal aspirates performed better than NPS and OPS [89].

For RSV, NSA provides higher specificity for diagnostic procedures than NPS [90]. To assess the spread of human metapneumovirus (hPMV) in newborns and children with respiratory infections, nasopharyngeal aspirates were taken and then examined by RT-PCR for the presence of hMPV [91].

For studying respiratory viruses using the NGS there are data to use from a group of 25–135 patients in studies, in other sources, coverage can reach hundreds of people. Sander van Boheemen (2019) examined 25 samples in his work [92] taken from children, there are examination data from 86, 110, and 135 patients.

Talia Kustin (2019) used 54 nasopharyngeal swabs to identify respiratory viruses by NGS sequencing using a Σ-Virocult M40-A Compliant kit (MWE, UK). Short-term storage was carried out at 4 C for 48 h, with a longer one, the samples were placed at −70 °C [93].

Collection and transport of respiratory samples require a cold chain to stabilize and prevent degradation of nucleic acids and false positive results. The sample is transported in a universal transport medium and stored at 2–8 °C for short-term or −70–80 °C for long-term storage. At the same time, repeated freezing and thawing of samples should be avoided [82,94,95].

The critical point at the stage of collecting and transporting samples is the preservation of the virus or genetic material; therefore, if the samples were not used immediately, they must be stored at a temperature of −80 °C. This condition is recommended in many scientific articles [96,97]. Thus, the final result depends on the conditions of sampling and the time of their use, it is necessary to minimize the time between the stages of sample preparation up to virus nucleic acid isolation [98].

#### 4.1.2. Virus Enrichment

The correct methodology for isolating DNA or RNA viruses from clinical samples is an important step in NGS, which will allow for maximum removal of host or bacterial DNA. Each researcher selects his method, which allows for obtaining sequences of the viruses with minimal losses. This stage is called the enrichment process. It may include various procedures to allow minimal loss of viruses while maximizing the removal of host or bacterial DNA.

The main steps of virus enrichment, before nucleic acid isolation, are usually methods based on the physical properties of virions—floating density, structure, size, etc. Common approaches to virus enrichment are filtration, centrifugation, nuclease treatment, virus concentration, and others. Various methodological approaches are presented in the literature, and they mainly depend on both the researcher and the technical capabilities of the laboratory.

J. Vibin (2018) presents the results of studies on the metagenomic detection of viruses in bird feces. In the work, various variants of sample enrichment were used. It has been shown that the optimal method for virus enrichment is to include an ultracentrifugation step for concentration and subsequent processing with nucleases [99].

Undoubtedly, ultracentrifugation is applicable for the simultaneous concentration and purification of the virus, which is based on the difference in size and floating density of virions and other related sample components. This is an effective way to precipitate viruses and eliminate other impurities. Simultaneously, ultracentrifugation is not available to all laboratories due to its high cost [100].

The commonly used step of sample enrichment is filtration through 0.2; 0.45; 0.8 µm filters as viruses typically range in size from 20–300 nm. Some researchers prefer filtering through 0.2 µm, others 0.45 µm or more [26,99,101]. As with any step involved in the sample enrichment process, filtration may be beneficial or may reduce virus recovery. Here, it is needed to carefully approach the size of the filter and the state of sample aggregation.

Sample treatment with nucleases [102] is also a frequently used approach by researchers to remove non-target DNA. The method is used to identify animal and plant viruses [103], which allows the elimination of nucleic acids, both DNA and RNA, while the viral genome remains protected in the viral capsid. At this stage, there is a risk of losing the target under study. It is not recommended for low-titer viruses if the viral nucleic acid is not in the capsid [100].

#### 4.1.3. Positive Selection Methods

Positive selection methods are used to enrich the samples with viral nucleic acids directly through virus-targeted probes, for example, using PCR, microarrays or virus capture methods, liquid phase hybridization, and NGS target methods.

NGS implies the use of target and sequence-independent nucleic acids and approaches. Targeted sequencing requires knowledge of the targets in question and includes amplicon sequencing or hybridization based on viral nucleic acid capture [104].

PCR is based on the use of primers targeted to related viruses or virus variants. The PCR-based approach is limited by its ability to detect only a specific taxonomical number of viruses due to multiplexity issues.

ARTIC Network primers are used for the amplification and sequencing of SARS-CoV-2. As new variants emerged and proliferated, the V3 primer set was found to poorly amplify some key mutations. J. Davis (2021) compared the results of sequencing samples with V3 and V4 primer sets and showed that the use of ARTIC V4 primers is critical for the accurate sequencing of the SARS-CoV-2 S gene. The lack of metadata describing the primer scheme used will negatively impact the subsequent use of publicly available SARS-CoV-2 sequencing reads and assembled genomes [105]. The use of ARTIC V4 primers for amplicons with a low reading depth significantly improves genome coverage of Alpha, Beta, Delta, Eta, and other variants of SARS-CoV-2 [106].

DNA microarrays are used to enrich the virus samples of any specific taxonomical group (family, species, and types) [107]. This technology has been used to detect some new viruses, including human cardio viruses, porcine circovirus, rhinovirus, and adenovirus. However, due to limited specificity, none of these methods is suitable for comprehensive vertebrate virus characteristics or virome analysis [100].

Sets of virus-specific probes have been developed for liquid-phase hybridization and selection of all known vertebrate viruses and virus variants. One of these approaches has been called virome capture sequencing (VirCapSeq) [108]. Technically, VirCapSeq consists of a specific oligonucleotide set (or probes) similar to oligonucleotides used in viral microarrays, in which a mixture with samples may hybridize or capture complementary nucleic acids. Despite the essential limitations of this method due to the continuous viral genome evolution, the developers have developed a set containing 2 million oligonucleotides permitting enlarging of the viral sequence range. This approach has high specificity comparable with other metagenomics methods and a high sensitivity comparable to real-time PCR.

Most capture-based methods use the mosaic array method when 80–120-mer DNA or RNA probes are used to cover the target genome’s length. Probes are attached to the previously fragmented genomic DNA/RNA, and the targets are eluted, ligated, and prepared for any specifically used sequencing platform [104]. This methodology uses the available commercial sets such as SureSelect XT, SeqCap Ez, VirCapSeq-VERT, and CATCH. These approaches, as the targeted amplification, limit their wide use for metagenomics studies related to new virus searches.

Another ViroCap selection approach was developed by Wylie et al. (2015) [109]. As VirCapSeq, ViroCap is also targeted to most types of viruses (from 34 viral families), which as it is known, infect the vertebrate. The use of ViroCap allowed a 296–674–fold increase in the number of virus readings compared to conventional metagenomic sequencing [100].

These approaches are shifted to the known virus sequence, and they may be effectively used to study mixed infections and ensure the sensitive characteristics of all viruses in clinical samples.

Amplicon sequencing includes the amplification of the target genome/genomes fragment using specific primers before the library preparation and sequencing. It is commonly used to study the diversity and structure of prokaryotic communities and is targeted to highly conserved rRNA genes [104]. As viruses have no universal conservative markers, this method limits its wide use for metagenomic studies related to new virus searches.

One of the most known methods is sequence-independent, single-primer amplification (SISPA). SISPA is the random priming method developed by Reyes and Kim in 1991. SISPA includes the targeted oligonucleotide ligation with the DNA molecule population. The total final sequence permits using of one chain of double-chain primer in the annealing, extension, and denaturation recycling with a high-precision polymerase. SISPA has been used to detect new viral agents, particularly in veterinary [110,111]. The disadvantage of this method is its many sequencing errors and uneven virus genome coverage [104].

Unlike targeted sequencing, sequence-independent sequencing does not require preliminary knowledge of the target genome(s). The information obtained during sequencing will contain the virus data and nucleotide sequence data remaining after sample enrichment, if appropriate, and this may be an advantage to obtaining any additional and new host DNA and/or available microbiota information. However, insufficient virus genome coverage may result in some virus detection sensitivity issues.

#### 4.1.4. Virus Enrichment Methods Based on Negative Exclusion

Based on the negative selection, these methods allow the removal of non-viral nucleic acids obtained from the host, reagents, and/or the environment. An example is the removal of rRNA from isolated nucleic acids and developed commercial kits, including rRNA “Ribo-Zero” by Illumina, “GLOBINclear” by Ambion, and others.

rRNA is the most common type of RNA in most cells. When sequencing samples, its presence leads to many non-viral reads, which significantly limits the number of relevant virus-associated sequences [104].

This approach is very effective in detecting viral RNA in samples such as serum and cerebrospinal fluid, where most of the host RNA is rRNA [83]. A limitation of this method is the lack of kits and reagents for rRNA elimination in other vertebrate species and this method is not promising when searching for viruses in cells and tissue samples, which usually contain large amounts of genomic DNA and transcripts [36].

#### 4.1.5. DNA/RNA Extraction Methods

The purification steps in the concentration and extraction process may not increase viral RNA, but the removal of unwanted nucleic acids may increase the ratio of virus reads and the quality of the resulting contigs [112].

A critical step in virus detection in clinical specimens is the efficient extraction of viral nucleic acids. The overall yield of viral nucleic acid from a clinical specimen depends on the volume of the specimen, the initial concentration of the virus, and the efficiency of the extraction method.

The extraction of viral nucleic acids is an important step in the molecular detection of viruses in clinical specimens [113,114]. While there are many manual and automated extraction methods, it is important to choose the most sensitive and reliable method for performing NGS. Studies have shown that the choice of extraction platform has a large impact on the reliability of diagnostic results [115,116], including bacteriome profiles, as well as on the detection of certain viruses with NGS.

Potential problems associated with extraction methods are cross-contamination of samples [117,118] and contamination with sequences present in the environment or molecular biology reagents (so-called “kits”) [119]. In viral NGS studies, these aspects represent a major challenge and must be accurately assessed [120,121]. The impact of nucleic acid extraction methods on human virome cross-infection is still poorly understood [122].

Marina Sabatier (2020) compared the sensitivity and contamination of samples and reagents of extraction methods used for viral NGS (eMAG; MagNA Pure 24, MP24, QIAamp RNA). QIAamp has a low proportion of virus reads for both clinical and spoof samples. Sample cross-contamination was higher with MP24 up to 36.09% of viral reads were mapped to spurious viruses in NTC (versus 1.53% and 1.45% for eMAG and QIAamp, respectively). The eMAG platform yielded a higher proportion of virus reads with limited reagent exposure and sample cross-contamination compared to the QIAamp and MP24 extractors [113].

J. Klenner (2017) compared commercially available nucleic acid extraction kits (QIAamp Viral RNA Mini Kit, QIAamp DNA Blood Mini Kit, QIAamp cador Pathogen Mini Kit, and QIAamp MinElute Virus Spin Kit). An evaluation of commercial nucleic acid extraction kits showed little difference in read numbers. It has been shown that a nucleic acid extraction kit that works well for PCR diagnostics can also be used for NGS diagnostics [115].

#### 4.1.6. Sequencing Quality Control

NGS is a very sensitive method and requires the use of quality controls to avoid false positive and false negative results. High-throughput sequencing technology requires high-purity processes and the inclusion of various controls. Sources of sequencing errors can be sample handling processes, contaminated reagents, consumables, or, more commonly, human error itself. To eliminate incorrect results, quality controls are introduced into NGS processes at various stages [123,124,125].

Studies within the Sequencing Quality Control (SEQC) project have shown that mistakes made by about 2–3% of the performer himself are of big importance these are labeling errors or incorrect pipetting [120,126,127].

For quality control, negative controls should be used to detect false positives. Their use compared with the samples being studied allows one to detect cross-contamination and conduct the correct interpretation of sequencing results [49,84,128].

In NGS it is important to use internal controls. Today, various commercial controls are available, including oligonucleotides, viruses, or genome fragments, a spike in RNA (SIRV), and the External RNA Control Consortium (ERCC) [104]. Their application is based on the introduction of the samples under study in the process of sample preparation. Simultaneously, according to [129,130], when using ERCC as controls, a large error was shown depending on the method used.

A. Bal (2018) used the bacteriophage MS2 (IQC) as an internal control, viral transport medium with MS2 was used as a positive external control (EQC) and only transport medium (NTC) was used to assess contamination [124]. Thus, the internal control MS2 indicates the purity of reagents and equipment. This control is also used as a control in PCR since this RNA also passed the quality control of cDNA synthesis. MS2 is a single-stranded RNA with a size of 3.5 kb, which allows it to be optimally used in processes for NGS and detection in the bioinformatic analysis [124,131,132].

### 4.2. NGS Platforms for Virome Studies

Initially, metagenomics was actively developed in the study of bacterial genomes and achieved tremendous success. However, a new direction in metagenomics, the virome has been actively developed. Virus studies using NGS methods are now at the peak of their development and technological approaches are being improved every time. Examples of applications are pathogen detection, including novel detection, species identification, and typing, detection of antibiotic resistance, virulence, and more [66,94,133].

With the development of NGS, its practical application is constantly expanding, especially in clinical virology in the diagnosis of new or previously undetected pathogens of infectious diseases [4,134,135]. It was shown that the sensitivity of the NGS method is comparable to that of the PCR method with increasing sequencing depth [5,66].

So, W.I. Lipkin (2013) in his works because of the bioinformatic analysis, revealed new viruses, such as rhabdovirus associated with acute hemorrhagic fever and cyclovirus found in the cerebrospinal fluid of patients [136,137]. Using the NGS method T. Kustin detected human parainfluenza 1 virus, human parainfluenza 4 virus, and influenza C among 54 patients [93].

In 91 samples of NFS by the NGS method were identified human rhinoviruses, enteroviruses, influenza A virus, coronavirus OC43 and respiratory syncytial virus (RSV) A, as well as rotavirus, torque teno virus, human papillomavirus, human betaherpesvirus 7, cyclovirus, vientovirus, gemycircularvirus and statovirus [138]. When examining NFS in 48 children, NGS revealed 11 RNA viruses, 4 DNA viruses, 4 bacterial species, and one fungus [97].

H. Mostafa (2020) in studies, when detecting SARS-CoV-2 by NGS in 500 patients, showed the possibility of diagnosing other infections and analyzing the respiratory microbiome [134].

Yi-Yi Qian (2021) showed that the sensitivity of NGS turned out to be higher than that of the traditional cultivation method, but in comparison with PCR, these indicators were lower [135]. Thorburn (2015) studied 89 nasopharyngeal swabs the sensitivity and specificity of the NGS method compared to Real-Time PCR were 78% and 80%, respectively [39]. So, the NGS technology as a diagnostic tool is still in the development stage, and approaches to its application are being improved every year.

Technically, NGS is run on various platforms, which can be divided by reading length into short-read and long-read. The short-read sequencing approaches fall into two categories: sequencing by ligation (SBL) and sequencing by synthesis (SBS).

In most approaches, SBL and SBS DNA are clonally amplified on a solid surface. The presence of many thousands of identical copies of a piece of DNA in a certain area ensures that the signal can be distinguished from background noise. Mass parallelization is also facilitated by the creation of many millions of individual SBL or SBS reaction centers, each with its own clonal DNA template. The sequencing platform can simultaneously collect information from many millions of reaction sites, thereby sequencing many millions of DNA molecules in parallel.

SBL technologies include Applied Biosystems/SOLiD and MGI/BGI/Complete Genomics. Sequencing by synthesis (SBS) is performed on the Illumina and Qiagen platforms.

Illumina offers a popular series of sequencing platforms–ISeq, MiSeq, MiniSeq, NextSeq, HiSeq, and NovaSeq. High throughput and low error rate (less than 1%) are the main reasons why this technology currently dominates the field of virology and beyond.

The very first NGS platform for studying viral metagenomics was Life Science/Roche 454, a pyrosequencing method. The 454 sequencing has been widely used to identify several new viruses and virome profiles from human and animal samples [139], including arboviruses [140], orbiviruses [29], arenaviruses [141], Lujo virus [142], astrovirus [143], gyroviruses [144], porcine bocaviruses [145], picornaviruses [146], rhabdoviruses [67], coronaviruses [147], gamma papillomavirus [148], and seadornavirus [149]. Most of these viruses have been identified in serum, respiratory, and feces samples.

Although this technology offered a higher yield than Sanger sequencing at a lower cost, this technology has been supplanted by other NGS technologies due to its high cost, errors in homopolymer regions, and low throughput.

Ion Torrent semiconductor sequencing technology with the Ion Proton and Ion S5 series sequencers which benefits from fast sequencing makes these sequencers particularly useful for targeted detection of viruses in clinical specimens, such as HIV [150], hepatitis B virus [151], HCV [152] and rapid genome sequencing of several viruses, including Tuscany virus [153], polyomavirus [154], porcine reproductive and respiratory syndrome virus [155], orthoreovirus [156], bluetongue virus [157], rotavirus [158], influenza virus [159]. This technology has been used to study the virome of skin [160], ticks [161], intestines in piglets [162] and seals [163].

Over the past few years, sequencing technologies have grown rapidly, introducing of third-generation sequencing (TGS) technologies such as Oxford Nanopore and PacBio platforms which are real-time single molecule sequencing (SMRT), that which reduces amplification bias and short reading problems. The reduction in cost and time presented by these sequencing methods is a valuable benefit.

TGS is considered the next revolution in sequencing technology. Sequencing of long sequences and speed, without PCR amplification, allows uniform coverage of the entire genome. This technology has also been used for virus sequencing [164,165,166]. Looking forward, future developments in TGS should focus on improving sequencing accuracy and high throughput.

### 4.3. Bioinformatics and NGS Data Analysis

An important stage in metagenomics is computer analysis or bioinformatics, the task is to process a big array of NGS data, which can be represented by sequences of the genomes of viruses, bacteria, humans, animals, and others.

When using early sequencing methods, sequences are usually classified using NCBI BLAST [167] against the NCBI (nt) database [168]. However, when using NGS data, it is necessary to process a much larger number of short (up to 300 bps) reads, for which homologous regions are not always available in databases and possible sequencing errors made by the sequencer must be considered.

Therefore, NGS needs specialized methods of analysis. Many biological information specialists have developed computational workflows for the analysis of viral metagenomes. Their publications describe many computer tools for taxonomic classification (Table 1). While these tools can be helpful, choosing the right workflow can be difficult, especially for less experienced users [169,170].

Bioinformatics involves the processing of sequencing data for checking the quality of reads, filtering sequences, and their identification. Some of the workflows of metagenomics have been tested and described in review articles [171,172,173,174,175].

There are specialized programs and online services for virus analysis, such as Viral MetaGenome Annotation Pipeline (VMGAP) [176,177], Viral Informatics Resource for Metagenomic Exploration (VIROME) [178] and Metavir 2 [179], DisCVR [180].

Additionally, there are cloud-deployed clinical metagenomic computing workflows such as SURPI (sequence-based ultra-rapid pathogen identification) [181] and CZ ID (IDseq) [182], for the detection and identification of pathogens.

The CosmosID program has been used to analyze the microbiome of various groups and quantify microorganisms [104,183,184].

Annotated data visualization programs are available for MEGAN, Pavian, Krona, PanViz, MetaViz, and Anvi’o. MEGAN and Pavian perform broad analyses but require specific inputs that make them less suitable for different workflows. PanViz, MetaViz, and Anvi’o are sharpened for the analysis of bacteria and are of little use for viruses. The available programs Geneious and CLC bio are paid for and require an expensive license [185,186,187,188,189].

To separate viral and non-viral sequences, vFams is used [190]. The VIP program is also used to identify viruses [35]. Virus-TAP, VirusSeker for BLAST-based virus identification with modules (VS-VIROME and VS-DISCOVERY), and SHIVER for de novo assembly [191,192,193].

In the age of NGS and bioinformatics, open, easily accessible, free, and globally distributed platforms for data analysis can significantly change the accessibility and quality of biomedical research. Baker et al. (2020) showed the possibility and importance of data exchange using the example of SARS-CoV-2. For example, for all virus genomic data, the Galaxy platform (https://usegalaxy.eu/ accessed on 18 November 2022) was used, which can be replicated using open-source tools by any researcher with an Internet connection.

The opportunities for such access allow for raising community awareness in the absence of primary data needed to respond to global emergencies such as the COVID-19 outbreak effectively, transparently, and reproducibly perform all analyzes on an equal footing.

Additionally, the publication emphasizes the problem of non-reproducibility of results that are published in scientific papers and, which cannot be completed again because the data are not shared or deliberately hidden. Thus, any researcher should be able to apply the same analytical procedures to their data and have access to all data analysis tools, including computing power and infrastructure [194].

## 5. Conclusions

The identification of viruses using PCR methods is challenging [195,196] and the genetic diversity of viruses can lead to a mismatch of probes and primer sequences, leading to incorrect PCR results [68]. These gaps can be addressed by NGS techniques. They are effective for the identification and genomic characterization of influenza A viruses (IAV) and other respiratory viruses [197].

Approaches based on NGS technologies may be a logical step as routine virus diagnostic on clinical specimens. Extending not only the range of virus detection but also providing an additional characterization of detected viruses, such as virus genotypes and subtypes. However, the efficacy and viability of using such methods for diagnostic purposes require further study.

Direct sequencing of viral genomes from clinical samples using NGS is fraught with difficulties, including the large presence of cell DNA, microbiota, and a limited amount of viral DNA/RNA [198]. Enrichment technologies for sample preparation have been developed to separate viruses from host cells [101].

The collection and storage of samples is a critical step in the detection of viruses by NGS technology, since viruses are weakly resistant and are subject to rapid destruction. Therefore, maintaining the cold chain and reducing the time between viral isolation steps is critical for obtaining results.

To obtain the maximum number of reads of the viruses various approaches are being used: physical enrichment, target enrichment, and negative exclusion. All approaches have been successfully used in sample preparation for NGS, while each has its advantages and disadvantages.

With sequence-independent RNA/DNA viruses, methods of physical enrichment are successfully used, which makes it possible to obtain the maximum number of target viruses at the final stage. Undoubtedly, the use of a high-speed ultracentrifuge makes it possible to obtain purified and concentrated viruses, but this does not apply to all laboratories. The main steps in the enrichment process are low-speed centrifugation, filtration through membranes with 0.2−0.8 µm, and destruction of host and bacterial DNA by nucleases. This method does not remove DNA impurities from prokaryotic and eukaryotic cells, but significantly reduces their presence [99,100].

Another effective method is an enrichment based on known DNA sequences. VirCapSeq showed high specificity comparable to other metagenomic methods and high sensitivity comparable to real-time PCR. At the same time, the absence of conserved regions in viruses does not allow it to be widely used in the search for new viruses. Good results are shown by the ISPA method for detecting new viruses, however, many sequencing errors and uneven coverage of virus genomes are observed [102,104].

Enrichment in negative selection is shown using rRNA removal as an example. This approach is effective in detecting viral RNA and eliminating host rRNA. The limitations of this method are the lack of rRNA kits for other vertebrate species [103].

An essential step is the isolation of viral DNA or RNA. It has been shown that the purification and concentration steps do not increase viral NA, but the removal of unwanted nucleic acids increases the number of virus reads and the quality of the resulting contigs. Studies to determine the optimal methods for isolating the nucleic acids of viruses for NGS sequencing have shown that the kits that are well recommended for PCR diagnostics are quite applicable for the preparation of NAs for NGS sequencing [115].

Currently, NGS platforms are widely used in the study of viral populations. NGS has been successfully used to detect new viruses, particularly in outbreaks of known and/or emerging diseases. The sensitivity of NGS was also shown, comparable to PCR in the case of an increase in the depth of sequencing. There are various NGS platforms. The widely adopted Illumina platform is characterized by high throughput and low error rates and dominates the field of virology. Ion Torrent technology is fast in sequencing and has been successfully used in the targeted detection of viruses in clinical specimens.

NGS sequencing technology and bioinformatic analysis are promising strategies for the rapid identification of pathogens in clinical and public health settings. This allows the characterization of known new pathogens that are unamenable to traditional testing, which is applicable in cases of rapid detection of the pathogen in critical cases.

However, it is currently unlikely that genome-based tools will soon be used in clinical diagnostic settings [195]. Its main barriers are cost-effectiveness, high throughput formats for clinical settings, and the need for investment in bioinformatics tools, databases, and data management. Further optimization of sequencing processes, standardization of processes, and common bioinformatic approaches must increase the accuracy, repeatability, and versatility of the technology.

The introduction of NGS into the diagnostic system will help advance research into respiratory infections. The use of NGS will allow the detection of viral and non-viral pathogens with simultaneous analysis of the genetic sequence. However, the effectiveness and feasibility of using NGS technology for diagnostic purposes require further study.

## Figures and Tables

**Figure 1 microorganisms-10-02327-f001:**
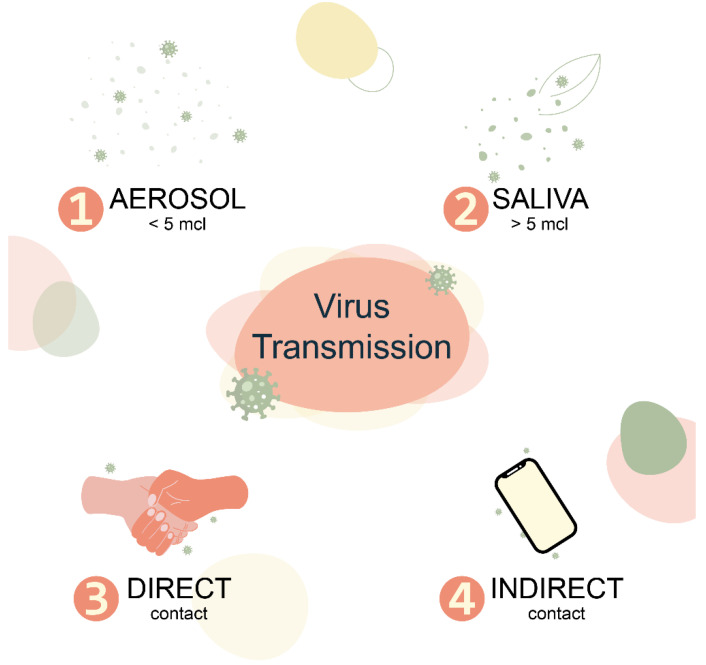
Most of the ways of transmission respiratory viruses.

**Figure 2 microorganisms-10-02327-f002:**
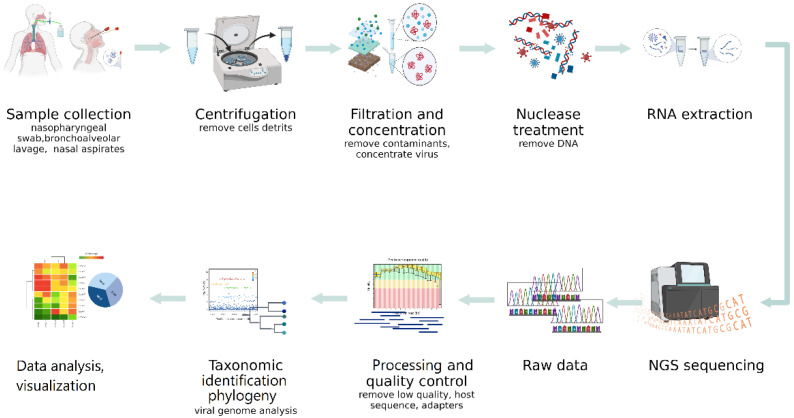
General scheme for NGS sequencing of respiratory viruses.

**Table 1 microorganisms-10-02327-t001:** List of software for NGS data analysis.

#	Software	Description	References
Processing of Sequencing Data			
1	Trimmomatic	It is a command line tool that can be used to trim and crop Illumina (FASTQ) data as well as to remove adapters	http://www.usadellab.org/cms/?page=trimmomatic (accessed on 14 February 2022)
2	PRINSEQ	It is a tool for quality control of metagenomic sequence data, includes functions for statistics, for trimming, filtering, and data reformatting	https://bioinformaticshome.com/tools/rna-seq/descriptions/PRINSEQ.html#:~:text=PRINSEQ%20is%20a%20tool%20for,quality%20measures%2C%20and%20tag%20sequences (accessed on 14 February 2022).
3	Cutadapt	Finds and removes adapter sequences, primers, poly-A tails and other types of unwanted sequence from high-throughput sequencing reads	https://cutadapt.readthedocs.io/en/stable/ (accessed on 14 February 2022)
4	MEGAN	For analysis of large metagenomic datasets. The set of DNA reads compared against databases of known sequences	https://bio.tools/megan (accessed on 14 February 2022)
Virus genome analysis			
5	Metavir 2	For a comprehensive analysis of assembled virome sequences	https://www.cd-genomics.com/bioinformatics-analysis-of-viral-metagenomic-sequencing.html (accessed on 14 February 2022)
6	MetaGeneAnnotator	It is a gene-finding program for prokaryote and phage	https://mybiosoftware.com/tag/metageneannotator (accessed on 14 February 2022)
7	VIP	For metagenomic identification of viral pathogens	https://github.com/keylabivdc/VIP (accessed on 14 February 2022)
8	VirusSeker	For eukaryotic virus discovery and composition analysis	https://mybiosoftware.com/tag/virusseeker (accessed on 14 February 2022)
9	VirusTAP	It is analysis tool for the viral genome	https://gph.niid.go.jp/cgi-bin/virustap/index.cgi (accessed on 14 February 2022)
Taxonomic classification			
10	DisCVR	For detection of known human viruses in clinical samples from high-throughput sequencing	https://bioinformatics.cvr.ac.uk/software/discvr/ (accessed on 14 February 2022)
11	KRAKEN	For assigning taxonomic labels to short DNA sequences, usually obtained through metagenomic studies	https://ccb.jhu.edu/software/kraken/ (accessed on 14 February 2022)
12	Bracken	It is a highly accurate statistical method for species of DNA. Bracken uses the taxonomy labels assigned by Kraken	https://ccb.jhu.edu/software/bracken/ (accessed on 14 February 2022)
13	Centrifuge	It is a microbial classification engine that enables rapid, accurate, and sensitive labeling of reads and quantification of species	https://ccb.jhu.edu/software/centrifuge/manual.shtml (accessed on 14 February 2022)
14	CLARK	For classification of short metagenomics reads at the genus/species using discriminative k-mers	http://clark.cs.ucr.edu/ (accessed on 14 February 2022)
15	VIROME	For classification of predicted open-reading frames (ORFs) from viral metagenomes	http://virome.dbi.udel.edu/ (accessed on 14 February 2022)
16	Taxonomer	For assigning taxonomy to sequencing reads from both clinical and environmental samples	https://taxonomer.iobio.io/#:~:text=Taxonomer%20is%20a%20kmer%2Dbased,meaningful%20timeframe%20(i.e.,%20minutes) (accessed on 14 February 2022)
Visualization tools			
17	Pavian	For exploring metagenomics classification results, with a special focus on infectious disease diagnosis. Analyze, display, and transform results from the Kraken and Centrifuge	https://ccb.jhu.edu/software/pavian/ (accessed on 16 February 2022)
18	Krona	It is an interactive visualization tool for exploring the composition of metagenomes	https://sourceforge.net/p/krona/home/krona/?version=27 (accessed on 16 February 2022)
19	PanViz	Visualization tool for investigating and understanding comparative microbial genomics data	https://github.com/thomasp85/PanViz (accessed on 16 February 2022)
20	MetaViz	For interactive exploratory data analysis of annotated microbiome taxonomic community profiles	https://mybiosoftware.com/tag/metaviz (accessed on 16 February 2022)
21	Anvi’o	It is an analysis and visualization platform that offers automated and human-guided characterization of microbial genomes in metagenomic assemblies	https://anvio.org/ (accessed on 16 February 2022)
22	Geneious	It is a DNA, RNA, and protein sequence alignment, assembly, and analysis software platform, integrating bioinformatics and molecular biology tools	https://www.geneious.com/ (accessed on 16 February 2022)
23	CLC bio	For bioinformatics analysis with graphical interface for building, managing and deploying analysis workflows	https://digitalinsights.qiagen.com/products-overview/discovery-insights-portfolio/analysis-and-visualization/qiagen-clc-genomics-workbench/ (accessed on 16 February 2022)

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
