# Peer review of "Next Generation Sequencing Approaches to Characterize the Respiratory Tract Virome"

_microorganisms, 2022, doi:10.3390/microorganisms10122327_

Round 1
Reviewer 1 Report
Sandybayev et al. reviewed the application of NGS in metagenomics studies. The focus is on both experimental solutions and computational workflows in NGS applications. In particular, experimental cautions (e.g., sampling scheme) were also included in the manuscript. The manuscript is overall well-written, and the topics covered are comprehensive. My major comments are on the conceptual level.
Major comments:
1. One of the first papers applying NGS to detect evolutionary changes in virus genomes is https://doi.org/10.1093/gbe/evp029. Include and discuss this paper.
2. It will be nice to include another paper from the Galaxy community (https://doi.org/10.1371/journal.ppat.1008643). This paper discusses how agile and effective responses to emerging pathogen threats can be made via open data and open analytics. Sandybayev et al. discussed the variation in performing NGS experiments and data analysis. This is exactly what the Galaxy community is advocating—reproducibility matters.
3. Line 371, one way to enrich virus genomes is via amplicon. The best example is the ARTIC scheme in SARS-CoV-2 detection. Virus genomes can also be captured using probes. In addition, LAMP is a good option.
4. Virus within-host heterogeneity is understudied. Please try to incorporate it in the manuscript.
Minor comments:
1. Line 86, "are likely to are central" → "are likely central".
2. Line 156, "SARS-CoV-19" → "SARS-CoV-2".
3. Line 244, "it must considered" → "it must be considered".
Author Response
Point 1. One of the first papers applying NGS to detect evolutionary changes in virus genomes is https://doi.org/10.1093/gbe/evp029. Include and discuss this paper.
Response 1:
The first studies on the use of NGS were conducted by B. Dickins (2009) to study evolutionary changes in the DNA of the phage UX174. A high degree of polymorphism in the phage genome was shown, including in cultures grown for several hours. The results indicated the need for more studies using different platforms to differentiate between instrument errors and mutational flicker data. This metagenomics approach has become the conceptual basis for further research into other viral infections.
Point 2. It will be nice to include another paper from the Galaxy community (https://doi.org/10.1371/journal.ppat.1008643). This paper discusses how agile and effective responses to emerging pathogen threats can be made via open data and open analytics. Sandybayev et al. discussed the variation in performing NGS experiments and data analysis. This is exactly what the Galaxy community is advocating—reproducibility matters.
Response 2:
In the age of NGS and bioinformatics, open, easily accessible, free, and globally distributed platforms for data analysis can significantly change the accessibility and quality of biomedical research. Baker et al. (2020) showed the possibility and importance of data exchange using the example of SARS-CoV-2. For example, for all virus genomic data, the Galaxy platform (https://usegalaxy.eu/) was used, which can be replicated using open-source tools by any researcher with an Internet connection.
The opportunities for such access allow for raising community awareness in the absence of primary data needed to respond to global emergencies such as the COVID-19 outbreak effectively, transparently, and reproducibly perform all analyzes on an equal footing.
Additionally, the publication emphasizes the problem of non-reproducibility of results that are published in scientific papers and, which cannot be re-done because the data are not shared or deliberately hidden. Thus, any researcher should be able to apply the same analytical procedures to their data and have access to all data analysis tools, including computing power and infrastructure.
Point 3. Line 371, one way to enrich virus genomes is via amplicon. The best example is the ARTIC scheme in SARS-CoV-2 detection. Virus genomes can also be captured using probes. In addition, LAMP is a good option.
Response 3:
ARTIC Network primers are used for the amplification and sequencing of SARS-CoV-2. As new variants emerged and proliferated, the V3 primer set was found to poorly amplify some key mutations. J. Davis (2021) compared the results of sequencing samples with V3 and V4 primer sets and showed that the use of ARTIC V4 primers is critical for the accurate sequencing of the SARS-CoV-2 S gene. The lack of metadata describing the primer scheme used will negatively impact the subsequent use of publicly available SARS-Cov-2 sequencing reads and assembled genomes.
The use of ARTIC V4 primers for amplicons with a low reading depth significantly improves genome coverage of Alpha, Beta, Delta, Eta, and other variants of SARS-CoV-2 (Lambisia, 2022).
Point 4. Virus within-host heterogeneity is understudied. Please try to incorporate it in the manuscript.
Response 4:
NGS methods are also used to study the genetic and phenotypic heterogeneity of viruses during replication in host cells, which is missed in conventional studies using consensus sequencing data.​​​​​​​ An example of such a study is Ikuyo Takayama (2021), who studied the genetic diversity of the A(H1N1)pdm09 virus using NGS in upper and lower respiratory tract samples from nine patients. Significant genetic heterogeneity was found in the samples, including 47 amino acid substitutions and one D222G/N substitution in the hemagglutinin was common to several patients. The authors note the need for such studies in order not to miss the potentially important mutations that occur during viral replication in the host, especially in patients with severe disease.
And also corrected minor comments.
Reviewer 2 Report
The manuscript described NGS approaches used for respiratory tract virome studies and clinical diagnosis in terms of NGS sequencing methods,platforms and data analysis via bioinformatics. The authors highlighted NGS methods with a focus on sampling, virus enrichment, positive selection methods, DNA/RNA extraction methods, sequencing quality control and discussed shortcomings and of each method. The manuscript is well structured and informative for NGS users. However, there are a few grammar errors and typos such as Line 256 to 258, Figure 2, Line 271, Line 297 to 298, lIne 305 to 307, Line 353.
Author Response
Point: The manuscript is well structured and informative for NGS users. However, there are a few grammar errors and typos such as Line 256 to 258, Figure 2, Line 271, Line 297 to 298, lIne 305 to 307, Line 353.
Response: Corrected grammatical errors and typos: line 256–258, fig. 2, line 271, line 297–298, line 305–307, line 353.